# Geometric Morphometric Analysis of Wing Shape to Identify Populations of *Apis mellifera* in Camagüey, Cuba

**DOI:** 10.3390/insects14030306

**Published:** 2023-03-22

**Authors:** Diego Masaquiza, Mario Octavio Ferrán, Santiago Guamán, Edwin Naranjo, Maritza Vaca, Lino Marcelo Curbelo, Amilcar Arenal

**Affiliations:** 1Sede Orellana, Escuela Superior Politécnica de Chimborazo, El Coca 220150, Ecuador; santiagoa.guaman@espoch.edu.ec; 2Centro de Estudios Para el Desarrollo de la Producción Animal, Universidad de Camagüey “Ignacio Agramonte Loynaz”, Camagüey 4650, Cuba; lino.curbelo@reduc.edu.cu; 3Facultad de Ciencias Pecuarias, Escuela Superior Politécnica de Chimborazo, Riobamba 060106, Ecuador; edwin.naranjo@espoch.edu.ec (E.N.); maritza.vaca@espoch.edu.ec (M.V.); 4Centro de Biología Molecular, Universidad de Camagüey “Ignacio Agramonte Loynaz”, Camagüey 74650, Cuba; amilcar.arenal@reduc.edu.cu

**Keywords:** geometric morphometrics, *Apis mellifera*, breeding lines, genetic diversity

## Abstract

**Simple Summary:**

Bees are one of the most important creatures on Earth because of their pollination processes, which contribute to food security and ecosystem maintenance. The practice of apiculture is regarded as secondary. Nonetheless, producers find this activity appealing due to the added value of its products. Researchers have been paying attention to the genetic erosion processes of pollinators for decades. To propose a conservation plan for these insects, identification studies from individuals from those locations must be carried out in the same way because different environmental conditions promote individuals with distinct characteristics that are harder to see with the naked eye.

**Abstract:**

A total of 45 *Apis mellifera* colonies were sampled from nine centers for rearing queens in the Camagüey province, Cuba. Wing geometric morphometric analysis was used to determine the ancestry and identify Africanization processes at different altitudes in managed honeybee populations on the island. A total of 350 reference wings were obtained from the pure subspecies: *Apis mellifera mellifera*, *Apis mellifera carnica*, *Apis mellifera ligustica*, *Apis mellifera caucasia*, *Apis mellifera iberiensis*, *Apis mellifera intermissa*, and *Apis mellifera scutellata* for the study. Our results showed that altitude influences wing shape; and that 96.0% (432) of the individuals were classified as Cuban hybrids, with a tendency to the formation of a new morphotype. In addition, a great similarity was found with the subspecies *Apis mellifera mellifera*, and it was confirmed that there is no Africanization due to the low presence of 0.44% (2) of this morphotype in the population under study. The greatest Mahalanobis distances were obtained for the comparisons between the center rearing of queens in the Camagüey province with the subspecies *A. m. scutellata* (D^2^ = 5.18); *A. m. caucasia* (D^2^ = 6.08); *A. m. ligustica* (D^2^ = 6.27); and *A. m. carnica* (D^2^ = 6.62). The well-defined pattern of wing shape produced by honeybee populations in Camagüey’s centers for queen rearing suggests a Cuban hybrid. Moreover, it is essential to note that the populations of bees under investigation lack Africanized morphotypes, indicating that Camagüey bees have not interacted with the African lineage.

## 1. Introduction

Bees are essential for the balance of ecosystems and play a critical role as pollinators, contributing to the reproduction and dispersal of most plant species, many of which are economically important [1]. Bees developed their activity as pollinators through a complex co-evolutionary process of these insects and flowering plants during the last hundred million years [2]. In the last decade, the dependence on agriculture for pollination services has increased [3].

The decline of species worldwide, caused by different factors and their synergy [4,5,6], significantly alarms researchers and producers. In this sense, the researchers have driven their efforts towards characterization studies, technological exploitation, and the sustainable use of the species [7].

In Cuba, the first colonies of *A. m. mellifera* (black or German bee) arrived and were introduced to the island in 1768 from Florida (US) to produce wax [8]. In the favorable environment of the territory, it expanded rapidly due to its hardiness, resistance to diseases, and high power of adaptation to different environments [9]. Subsequently, in 1904, the subspecies *A. m. ligustica* (yellow or Italian bee) was introduced, corresponding to a different lineage, and a crossbreeding occurred between them [8].

From 1960 to 1970, queens of *A. m. caucasia* were imported from southern Russia. Small introductions of *A. m. carnica* and *A. m. caucasia* have also been reported [8]. The last documented introduction was in 1985, after the importation of Italian-bred queens from New Zealand. The sanitary authorities implemented a strict prohibition of biological material on beekeeping importation [10], which is still in force today.

Free crossing of the introduced bee breeds produced the bee existing today in the country, which underwent an inevitable process of hybridization and adaptation. In addition to the effect of spontaneous selection actions carried out by beekeepers, the current “Cuban Creole” bee is appreciated for its diverse coloration and moderate defensive behavior. Depending on the behavior of climatological variables, the floristic richness of beekeeping interest, and its management, the Cuban bees achieve productivity averages among the best in the world. Despite the 60-year ban on the introduction of new bees in Cuba, a recent article demonstrates that 20% of the haplotypes correspond to African origin. However, the study used only 34 samples [11].

Since ancient times, several methods have been used to identify and classify *A. mellifera*, such as traditional morphometry [12,13]. The need to identify *A. mellifera* hybrids resulting from interbreeding between the various subspecies continues to improve with specific methods such as the Africanized Bee Rapid Identification System (FABIS) for the preliminary identification of suspected Africanized bees [14].

The Universal System for Detecting Africanized Identification (USDA-ID), which is necessary to declare official cases of Africanization [15], is a laborious process because twenty-five mounted parts of each specimen are needed. The Automatic Bee Identification System (ABIS), which takes two minutes for each sample and uses linear discriminant functions to identify colonies, and is used to compare a digital image of the specimen’s anterior part with wing-vein plots. In addition, 99.2% of the hives were correctly identified by ABIS when using Africanized bee samples [16].

Geometric morphometry (GM) is a relatively recent approach, and it has had an enhancement due to its accuracy of analysis [17] for discrimination between species, subspecies, and hybrids [18]. GM shows a better power and an approach that describes shapes using landmarks [19]. The method uses Cartesian coordinates of anatomical reference points located at the intersections of the wing veins, also called landmarks or homologous points. They are specific and located according to some criteria (homology, shape coverage, and coplanarity) on a biological structure or an image, whose purpose is to extract information from the geometry shape for particular comparative purposes [20].

GM employs a comprehensive statistical analysis to extract spatial information of morphological structures, increasing the accuracy concerning traditional morphometry [21]. The method allows a critical analysis of the morphometric variation of a given structure in organisms of various sizes [22]. This method discriminated against 24 known subspecies of honey bees in Europe [17]; meanwhile, Oleksa and Tofilski [23] indicated that GM provides similar discrimination as microsatellites. Geometric morphometrics is less expensive and easier to use than molecular methods, making it an alternative or complementary tool for identifying honeybee lineages, subspecies, and even intra-subspecies structures [16].

Currently, there are locally adapted populations in Cuba, with a mosaic of the distinctive morphological characters against the originally introduced subspecies [10,11]. The National Bee Research Institute runs the queen selection program in Cuba. The program aims to achieve a genetic response in all the characteristics of interest (honey production, hygienic behavior, *Varroa* infection, aggressive behaviour, etc.), limit the expansion of inbreeding, and preserve the genetic background. Every province has queen-rearing centers for sustainable genetic progress, where various phenotypic markers are evaluated. The program lacks tools to evaluate its effectiveness and GM is a powerful tool to evaluate honeybees. Therefore, this paper aimed to characterize *Apis mellifera* bees from the centers for rearing queens located in the province of Camagüey, Cuba, using geometric wing morphometry.

## 2. Materials and Methods

### 2.1. Geographical Location

The research was conducted in 2018 in nine centers for rearing queens of the municipalities Vertientes, Florida, Esmeralda, Camagüey, Najasa, Minas, Jimaguayú, Sibanicú, and Guáimaro in the province of Camagüey (Figure 1).

### 2.2. Sampling

Five hives were sampled in each center for rearing queens, and ten worker bees were randomly collected from the central combs of the brood chamber. The samples were kept in tubes with 96% ethanol and subsequently stored at −20 °C until their analysis in the molecular biology laboratory belonging to the Faculty of Agricultural Sciences of the University of Camagüey.

### 2.3. Geometric Morphometrics

The left anterior wings of 450 workers were dissected and put on microscope slides and scanned with a PlusteK OpticFilm 8100 (7200 dpi). On the images obtained, 19 landmarks were manually marked on the venal intersections of the wing [16] with the tpsDig2 v2.3 software, State University of New York, Stony Brook, NY, USA (Figure 2). The tps files were prepared using the tpsUtil v1.46 software, State University of New York, Stony Brook, NY, USA [16].

Additionally, we included fifty images of the left anterior wing from each subspecies (*A. m. carnica*, *A. m. caucasia*, *A. m. ligustica*, *A. m. mellifera*, *A. m. intermissa*, *A. m. iberiensis*) with previous evidence of their presence in Cuba [10,11]. *A. m. scutellata* was included to identify the possible presence of Africanization in the population under study. All images were obtained from The Morphometric Bee Data Bank in Oberursel, Germany.

### 2.4. Statistical Analyses

We first produced a Procrustes distance fit to eliminate variations caused by differences in size, position, and orientation of the wings. The residuals of this regression were used as “size-free” variables. The superimposed coordinates were projected in space [24]. These data were used as inputs in a principal component analysis (PCA), canonical variate analysis (CVA), and discriminant function analysis (DFA).

A cross-validation test verified the reliability of the data, and a permutation test was carried out for all pairwise tests. The data obtained were analyzed with three levels of classification: first, the sampling was considered as a whole in search of specific groups; and at the subspecies level, which was contrasted with the various patterns of the bees under study and the pure subspecies that entered the country, to determine their ancestors and also to identify Africanization processes [16].

The differences in wing shapes between the different groups of bees were observed using the scatter diagrams of the specimens along the first two canonical axes. We also calculated the Mahalanobis square distances between the centroids of the groups’ distribution and built a phenogram of morphological proximity based on the neighbor-joining algorithm using MEGA 7.0 [25].

## 3. Results

The PCA allowed observing the variation within and between the different centers for rearing queens and identifying the main characteristics of variations with the deformation graph (Figure 3), showing differences in the size and shape of the wing. The essential landmarks responsible for the variation in the non-affine or non-uniform components were 6, 7, 16, 19 in PC1 and 6, 13, 14, 16 in PC2 (variance 0.00074605).

The PCA generated 34 measures of relative deformations. The first five components are responsible for 60% of the total variance, and PC 1 and PC 2 represented 16.94% and 15.53%, respectively (Figure 3).

When comparing the seven pure subspecies and the bees of the centers for rearing queens, the results of the CVA indicated that the first five variables account for more than 80% of the total variation of the shape and size characteristics of the *A. m.* wing. The most significant difference can be seen in the canonical variates one and two representing 50% and 15.6% of variation in the samples, respectively (Figure 4).

The scatter diagram showed the separation into groups and the similarity between the populations of the subspecies *A. m. mellifera* and the bees from the centers for rearing queens, although there is a tendency for a new morphotype with particular characteristics. The results indicate that the wing venation pattern differs significantly between subspecies. It allows discrimination between them and provides sufficient information for new morphotypes.

When carrying out the discriminant function analysis, 97.29% of the individuals were classified into their respective groups. However, when applying the cross-validation test, 96% of them were correctly identified (Table 1). Of the samples, 400 individuals (92.67%) were classified as Cuban hybrids; 19 (4.40%) *A. m. mellifera* (Lineage M); 1 (0.23%) *A. m. ligustica* (Lineage C); 1 (0.23%) *A. m. carnica* (Lineage C); 2 (0.46%) *A. m. caucasia* (Lineage O); 1 (0.23%) *A. m. iberiensis* (Lineage M); 7 (1.62%) *A. m. intermissa* (Lineage A) and 2 (0.46%) identified as *A. m. scutellata* Lineage (A).

Figure 5 shows vectors and grids after Procrustes distance and the Mahalanobis distance analyses. We observed lower grid deformation with the highest similarity between the subspecies and hybrids. The plate spline showed that the highest differences were seen in pairs with C honeybee lineages.

The Mahalanobis distances were closest for *A. m. mellifera* (D^2^ = 3.77), and showed greater distances concerning the subspecies *A. m. scutellata* (D^2^ = 5.18); *A. m. caucasia* (D^2^ = 6.08); *A. m. ligustica* (D^2^ = 6.27); *A. m. carnica* (D^2^ = 6.62); *A. m. iberiensis* (D^2^ = 4.92); and *A. m. intermissa* (D^2^ = 4.3). The UPGMA analysis resulted in four main clusters, locating a Phenon line at 2.68 (Figure 6). The first includes individuals of the A lineage (*A. m. scutellata*), the second, members of the lineage O (*A. m. caucasia*); the third, members of the C lineage (*A. m. ligustica* and *A. m. carnica*), and the fourth is represented by two sub-groups: populations of the M lineage (*A. m. mellifera*) and Cuban hybrids.

## 4. Discussion

The results obtained in the analysis with geometric morphometry indicate that the left forewing provides enough information to distinguish honeybee morphotypes from different geographic regions. Based on the results obtained in this research, it can be affirmed that the honeybee populations of the centers for rearing queens of Camagüey are structured mainly from hybrids with the most significant similarity with the subspecies *A. m. mellifera*, which constitutes the first report to indicate such a discovery. However, it is crucial to highlight that, according to the distribution reached by Cuban hybrids according to the subspecies analysis, there is evidence for the tendency to a new morphotype with particular characteristics.

UPGMA clustering based on Mahalanobis distances also supported discriminant function analysis of subspecies. *A. m. intermissa* and *A. m. scutellata* were members of lineage A’s subgroup. The existence of four honeybee lineages is strongly supported by the results of UPGMA clustering of geometric morphometric data. Similar findings were obtained by [17]. The authors were able to determine that a geometric morphometric method consistently distinguished between 24 honeybee subspecies and four lineages.

Furthermore, Mahalanobis distance indicates that the subspecies *A. m. scutellata* is distant from the Cuban hybrids. This is significant and suggests that Africanization is not, or is only minimally present in the bees from our study. García et al. [11], using cytochrome b sequence analysis, suggested a common ancestry between current Cuban and European bee populations. Several studies indicate consistency between the results of morphometric and genetic methods [26,27]. Single nucleotide polymorphism panels have also been developed to determine the degree of Africanization and ancestry in New World and Australian honey bees [28]. Notwithstanding, for *A. mellifera* subspecies, whose populations have been examined, genetic information at this degree of detail is still deficient.

Tofilski [29] demonstrated that geometric morphometrics performed marginally better than standard morphometry at distinguishing three honeybee subspecies—*A. m. mellifera*, *A. m. carnica*, *and A. m. caucasia*. Probably the characteristics of the rest of the subspecies introduced into the country (*A. m. ligustica*, *A. m. carnica*, and *A. m. caucasia*) were dissolved with the different crossing processes, which occurred spontaneously or directed by man, furthermore to the influence of microevolutionary processes, such as the founder effect, local adaptation, and bottleneck. *A. m. mellifera* was introduced into Cuba in 1763 [8], leading to its wide distribution throughout the country, favored by its characteristics of rusticity, resistance, and adaptation to different environments, which allowed for its rapid spread.

The differences between lineages or subspecies could be plotted as vectors and deformation grids thanks to geometric morphometric analysis. The landmark regions that contribute the most to discrimination are identified by the deformation grid. The deformation grids showed the differences between the lineages, and pairs with C honeybee lineages showed the most differences. It was confirmed, by UPGMA analysis, that the C lineage was the first cluster to separate from the rest of the lineage.

Benítez et al. [30], using geometric morphometry of the wing, found that there was a slight variation in the configuration of the landmarks in populations of the centers for rearing queens in the Granma province in eastern Cuba without achieving evident discrimination between groups. The article lacks a comparison of endogenous populations with honeybee pure populations. In another study focused in the western part of the island, the authors did not find a high degree of fluctuating asymmetry in Sancti Spiritus and Cienfuegos [10], through measurements of morphometric characters indicated that the hives belonging to the center for rearing queens in the Mayabeque province correspond to the “European” racial status. However, these studies have a limit in distinguishing which of the European subspecies that entered the country is most closely related to the Cuban populations.

Secondary overlaps and minor Mahalanobis distances were observed between *A. m. mellifera* and *A. m. scutellata*, respecting the rest of the subspecies. Whitfield et al. [31] confirmed that this approach between the two subspecies is because they share a common ancestor. Miguel et al. [32] identified the relationship between the Canary and Iberian bees and the M lineage; the D^2^ value was 3.87, which confirmed that the towns of La Palma and Tenerife were the closest. In Neotropical Nicaragua, mitotype A4 dominates at higher altitudes, indicating a high degree of Africanization in *A. mellifera* colonies. There were 21 of the mitotype A4 *A. m. scutellata* [33]. A recent comprehensive study [34] that included 500 colonies collected from the five beekeeping regions of Mexico provided a clearer picture of the current genotypic makeup of honeybees in the country. African mitotypes were found in 51.5% of the colonies that were sampled. The tropical beekeeping region of the Gulf coast had the highest frequency of African mitotypes (69.8%), followed by the Yucatan Peninsula (63.8%) and the Pacific coast (63.1%). The Northern region, where European mitotypes predominate, had the lowest frequency of African mitotypes (24.9 percent).

Francoy et al. [35] stated that the wing venation patterns of Africanized bees are highly influenced by this subspecies and demonstrated that the wing venation pattern of Africanized bees is genetically dominant over that of Italian bees. Concerning this aspect, Benítez et al. [30] ensure that the existing geographic variation between the localities where the centers for rearing queens are located has a determining function. Although the heights are small, some elements can be natural physical barriers and cause the isolation of some populations. (Oyerinde et al. [36]) found two different morphotypes when investigating the bee populations belonging to five agroecological zones in the savanna vegetation in Nigeria.

## 5. Conclusions

Honeybee populations in the centers for rearing queens in Camagüey produce a well-defined pattern of a wing shape, suggesting a Cuban hybrid. The similarities between individuals were influenced by beekeeping practices such as transhumance, but mainly by the massive trade of queens from a small group of selected breeders. However, it is essential to note that the bee populations under study have a low presence of Africanized morphotypes, suggesting the lack of contact of Camagüey bees with the African lineage.

## Figures and Tables

**Figure 1 insects-14-00306-f001:**
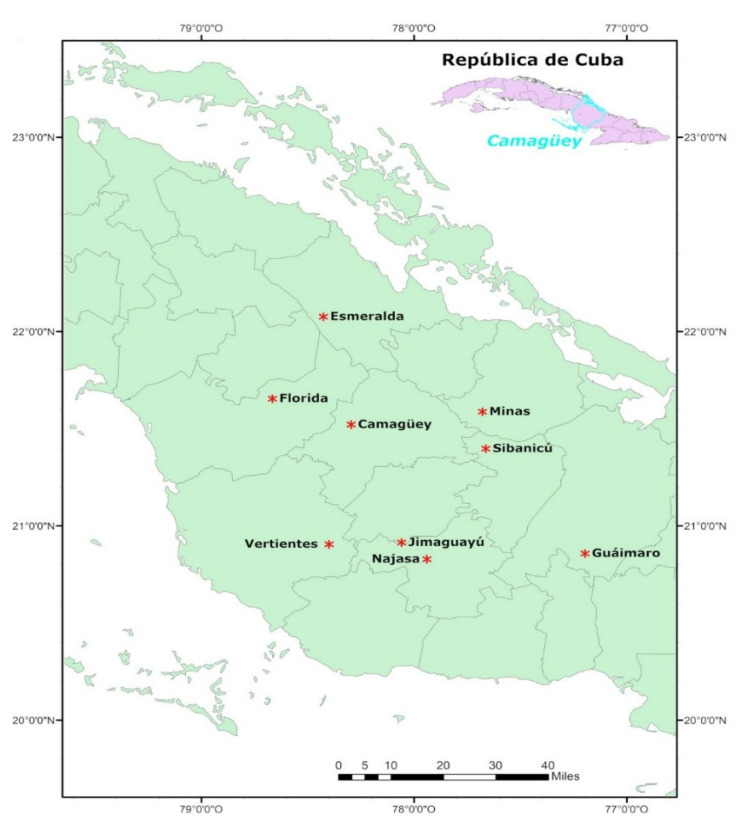
Map of Camagüey province with the geographic location of the centers for rearing queens. (*, Camagüey municipalities).

**Figure 2 insects-14-00306-f002:**
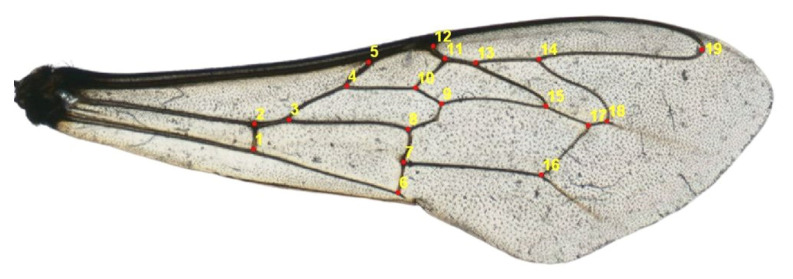
Location of 19 landmarks at venal intersections of the left forewing of *Apis mellifera* worker.

**Figure 3 insects-14-00306-f003:**
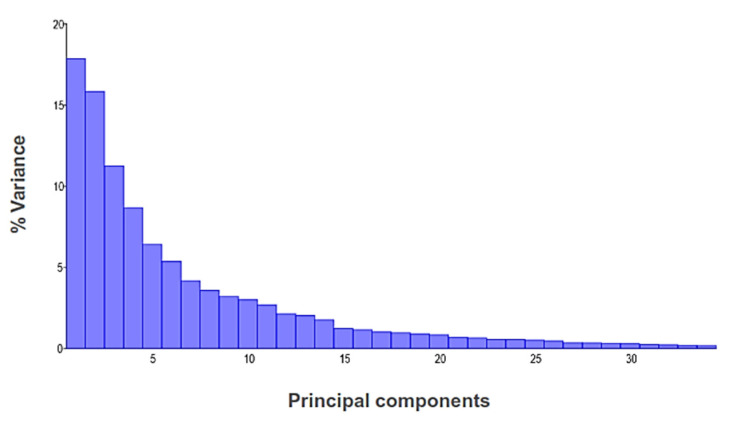
Eigenvalue showing the relative percentage of principal components among populations of *Apis mellifera*.

**Figure 4 insects-14-00306-f004:**
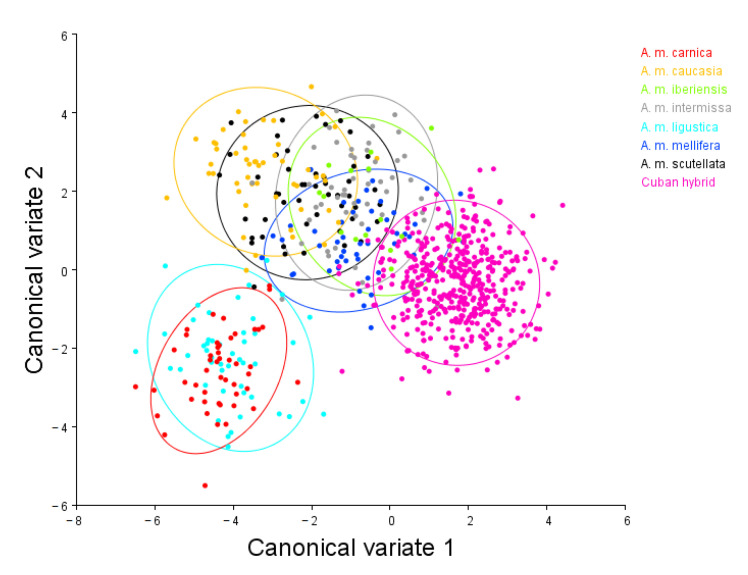
Scatter diagram according to the landmarks of the left forewing of *Apis mellifera* from the centers for rearing queens. CVA: considering pure subspecies, Red: *A. m. carnica*, Orange: *A. m. caucasia*, light green: *A. m. iberiensis*, gray: *A. m. intermissa*, light blue: *A. m. ligustica*, blue: *A. m. mellifera*, Black: *A. m. scutellate*, and centers for rearing queens under study. Ellipses are drawn with 90% probability.

**Figure 5 insects-14-00306-f005:**
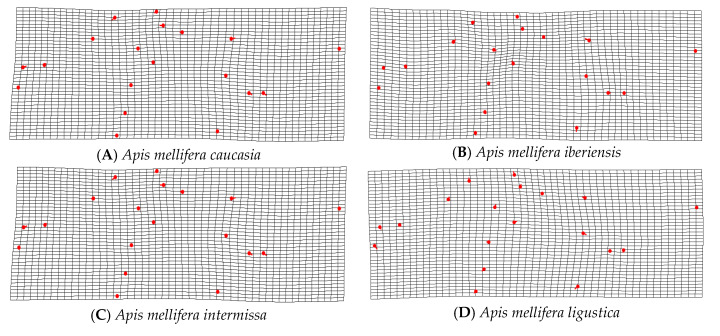
Comparison of deformation grids of bees from the centers for rearing queens with.

**Figure 6 insects-14-00306-f006:**
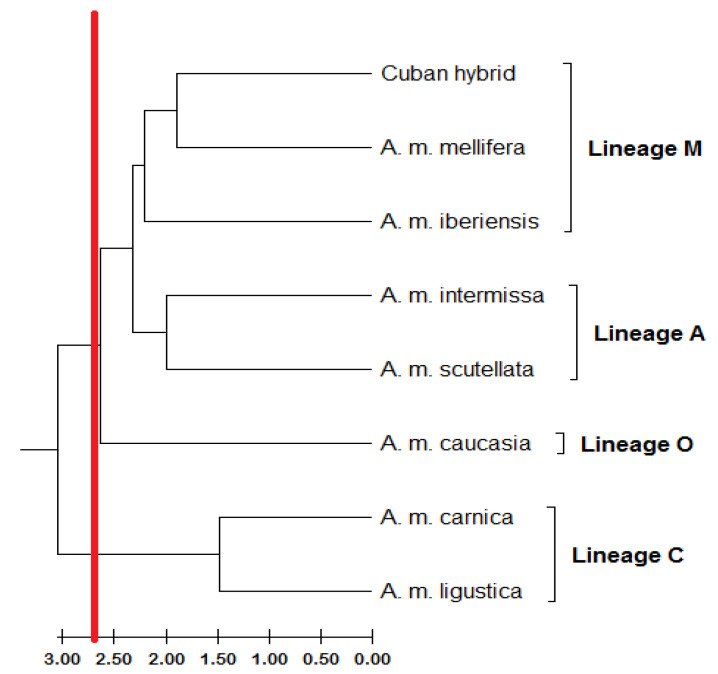
UPGMA phenogram showing the geometric morphometric relationship among honey bee subspecies based on Mahalanobis distances computed from the clusters of the pure subspecies of *Apis mellifera* and the centers for rearing queens. Method: Pairwise distance. Bee lineages: M, C, O, A. Subspecies: *A. m. iberiensis*, *A. m. intermisa*, *A. m. mellifera*, *A. m. carnica*, *A. m. ligustica*, *A. m. caucasia*, *A. m. scutellata*. Red line = phenon.

**Table 1 insects-14-00306-t001:** Frequency of bees (wings) of the centers for rearing queens based on the pure subspecies of *Apis mellifera*.

Known Classification	N (wings)	*A. m. carnica*	*A. m. ligustica*	*A. m. mellifera*	*A. m. scutellata*	*A. m. caucasia*	*A. m. iberiensis*	*A. m. intermissa*	Cuban Hybrid
*A. m. carnica*	50	100							0
*A. m. ligustica*	50		100						0
*A. m. mellifera*	50			96					4
*A. m. scutellata*	50				98				2
*A. m. caucasia*	50					100			0
*A. m. iberiensis*	50						98		2
*A. m. intermissa*	50							98	2
Cuban hybrid	432	0.23	0.23	4.40	0.46	0.46	0.23	1.62	92.67

## Data Availability

Research data, including coordinates of the reference points and wing images, are shared in the public access repository “Zenodo”: Masaquiza Moposita Diego Armando, Ferran Mario, & Arenal Cruz Amilcar. (2023). Collection of images of honeybee wings (Apis mellifera) from Cuba. Zenodo. https://doi.org/10.5281/zenodo.7662013, accessed on 20 March 2023.

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
