# Peer review of "Geometric Morphometric Analysis of Wing Shape to Identify Populations of Apis mellifera in Camagüey, Cuba"

_insects, 2023, doi:10.3390/insects14030306_

Round 1
Reviewer 1 Report
The authors investigated the morphometry of the wings from honeybee sample in Cuba.
Next time, it would be nice to have each line numbered...
I understand the choice to select the left anterior wing from 5 subspecies based on historical evidences but why not all of them ? 100 years ago, the Apis mellifera importation to Cuba was controlled ? Quid of the other subspecies such as Apis mellifera adansonii or Apis mellifera unicolor, maybe other subspecies could better explain the new morphotype that you observe in your result ?
There is also another analysis to try: what is the spatial autocorrelation of your sample, maybe more than altitude the morpholical differences in the wing is explained by the geographic position of your data.
I think that the conclusion of your paper could not be supported by your data as there are other honeybee subspecies to test, even it is strongly possible that you have an hybrid subspecies.
So, I think that you must let the possibilities in the discussion and conclusion that you have not tested all the possibilities.
What are the Genetic Centers of Queen production ? You need to detail the origin of the word GCQ in your manuscript.
Figure 1 need to be improve, it is difficult to see on which part of the island the samples are collected. Moreover, a scale must be added
Figure 2 A scale must be added
What is a Procrustes ? Why do you group them by altitude, it was not introduce in your research question ?
Figure 5 : Please add the % of Variance for each axis.
I do not understand the purpose of Figure 3 and 8
In the conclusion, in what aspect the honeybee population differ by regions ? It is not my interpretation of Figure 5.
Author Response
"Please see the attachment."

Reviewer 2 Report
The manuscript presents morphometric data which confirm that in Cuba there are honey bees of European origin (from lineage M). It is interesting that Africanized bees are not present there.
Similar results were published earlier. There are also some other week points mentioned below. However, I would recommend acceptance of the manuscript for publication if Authors share their research data including landmark coordinates and wing images. Data sharing is encouraged by the instructions for Authors. The data should be shared in a publicly accessible repository for example Zenodo.
I started to share my data and encourage others to reuse it, for example:
Oleksa, A., Căuia, E., Siceanu, A., Puškadija, Z., Kovačić, M., Pinto, M.A., Rodrigues, P.J., Hatjina, F., Charistos, L., Bouga, M., Prešern, J., Kandemir, I., Rašić, S., Kusza, S., & Tofilski, A. (2022). Collection of wing images for conservation of honey bees (Apis mellifera) biodiversity in Europe [Data set]. Zenodo. https://doi.org/10.5281/zenodo.7244070
In the manuscript there should be added information about another studies which show or suggest lack of Africanization in Cuba:
García, C. A. Y., Luis, A. R., Perez Pineiro, A., Perez Morfi, A., Invernizzi, C., & Tomasco, I. H. (2021). Cuban honey bees: significant differentiation from European honey bees in incomplete isolation. Journal of Apicultural Research, 60(3), 375-384.
Pérez, A., & Demedio, J. (2017). Racial status and index of hive (Apis mellifera L.) infestation by Varroa destructor (Anderson and Trueman) in Mayabeque, Cuba. Cuban Journal of Agricultural Science, 51(2).
Less relevant but also interesting to me is information that the honeybees from Cuba tolerate Varroa
Luis, A. R., Grindrod, I., Webb, G., Piñeiro, A. P., & Martin, S. J. (2022). Recapping and mite removal behaviour in Cuba: home to the world’s largest population of Varroa-resistant European honeybees. Scientific Reports, 12(1), 15597.
It is not clear what is the sample size. In methods there is information that 5 hives were used, 10 workers per hive. But later there is information about 450 individuals. It is important information if there was 5 or 45 colonies.
The data should be averaged within colonies.
Colonies should be analysed not workers especial in case of comparison with other subspecies. All workers from one colony belong to the same subspecies.
An important problem of the manuscript is lack of statistical tests for differences between groups.
Some information presented in the manuscript indicate lack of understanding of the statistical methods used. For example: "However, according to altitude (50-70; 71-90 and 91 at more than 100 m.a.s.l.), the AVC showed the first and second canonical variables explained 100 % of the total varia-tion of the sample."
As I understand, if there is 3 groups the first 2 CV always explain 100% of variance. It should be CVA not AVC.
In the end of discussion there a random collection of references not related to the subject of the study. It would be more interesting to review information about Africanization of neighbouring countries.
Minor suggestions
Line numbers should be added.
It should be stressed that Florida is a town in Cuba and not state in USA
More corrections can be found in the attached pdf file.

Author Response
"Please see the attachment."

Reviewer 3 Report
insects-2188776 Reviewer comments
Manuscript insects-2188776: Geometric morphometry analysis of the wing to identify populations of Apis mellifera in Camagüey, Cuba.
The manuscript is very interesting. The authors based on wing geometric morphometric determined the ancestry and identified Africanization processes at different altitudes in managed bee populations on the island. 250 reference wings were obtained from the pure subspecies: Apis mellifera mellifera, Apis mellifera carnica, Apis mellifera ligustica, Apis mellifera caucasica and Apis mellifera scutellata for the study. Our results show that altitude influences wing shape; and that 96.0 % (432) of the individuals were classified as Cuban hybrids, with a tendency to the formation of a new morphotype. In addition a great similarity was found with the subspecies Apis mellifera mellifera, and it was confirmed that there is no Africanization due to the low presence of 0.44 % (2) of this morphotype in the population under study.
The uniqueness of the text is 90% by antiplagiarism.net
The English is good but there are misspellings.
The experimental and statistical methods are correct.
There are some mistakes and comments:
Line 30 - adittion - should be - addition.
Lines 43-45 - after the sentence - Bees are essential for the balance of ecosystems and play a critical role as pollinators, contributing to the reproduction and dispersal of most plant species, many of which are economically important. - add citation (Ilyasov et al., 2020).
Add to the References - Ilyasov, R.A.; Lee, M.-l.; Takahashi, J.-i.; Kwon, H.W.; Nikolenko, A.G. A revision of subspecies structure of western honey bee Apis mellifera. Saudi J. Biol. Sci. 2020, 27, 3615-3621, doi:10.1016/j.sjbs.2020.08.001.
Line 59 - Apis mellifera caucasica - should be - Apis mellifera caucasia (according to current taxonomy names)
Line 60 - use full name Apis mellifera only in first time in the text and further use abbreviation A. m.
Line 82 - after the sentence - Geometric morphometry (GM) is a relatively recent approach, and it has had an enhancement due to its accuracy of analysis [15] for discrimination between species, subspecies, and hybrids. - add citation (García et al., 2022).
Add to the References - García, C.A.Y.; Rodrigues, P.J.; Tofilski, A.; Ilyasov, R.; Elen, D.; McCormak, G.P.; Oleksa, A.; Henriques, D.; Kartashev, A.; Bargain, C., et al. Using the Software Deepwings© to Classify Honey Bees Across Europe Through Wing Geometric Morphometrics. Insects 2022, 13, 1132, doi:10.3390/insects13121132.
Line 212 - Hibrids - should be - Hybrids.
Line 224 - A. m. scuttelata - should be - A. m. scutellata.
Line 266 - Apis mellifera scuttelata - should be - Apis mellifera scutellata.
Discussion part is weak, please add more discussion.
Please improve the manuscript according to the above comments.
Author Response
"Please see the attachment."

Round 2
Reviewer 1 Report
When you use an acronym at line 90 for "Geometric morphometry (GM)" please re-use it in the text e.g at line 92, check it throughout the text
Author Response
Thank you very much for your time and your recommendations
Reviewer 2 Report
The manuscript was corrected and the wing images were provided.
In the attached pdf file there are few minor corrections.

Author Response
Thank you very much for your time and your recommendations to the MS.
We have reviewed carefully and made all the grammar corrections. We also have included the information referring to table 1